# Epithelial-Mesenchymal Transition and MicroRNAs in Colorectal Cancer Chemoresistance to FOLFOX

**DOI:** 10.3390/pharmaceutics13010075

**Published:** 2021-01-08

**Authors:** Paula I. Escalante, Luis A. Quiñones, Héctor R. Contreras

**Affiliations:** 1Laboratory of Chemical Carcinogenesis and Pharmacogenetics (CQF), Department of Basic and Clinical Oncology (DOBC), Faculty of Medicine, University of Chile, 8500000 Santiago, Chile; paula.escalantee@gmail.com; 2Laboratory of Cellular and Molecular Oncology (LOCYM), Department of Basic and Clinical Oncology (DOBC), Faculty of Medicine, University of Chile, 8380453 Santiago, Chile; 3Latin American Network for the Implementation and Validation of Pharmacogenomic Clinical Guidelines (RELIVAF-CYTED), 28015 Madrid, Spain

**Keywords:** microRNA, epithelial-mesenchymal transition, 5-fluorouracil, oxaliplatin, FOLFOX, chemoresistance, pharmacogenetics, pharmacoepigenetics, EMT-transcription factors, biomarker

## Abstract

The FOLFOX scheme, based on the association of 5-fluorouracil and oxaliplatin, is the most frequently indicated chemotherapy scheme for patients diagnosed with metastatic colorectal cancer. Nevertheless, development of chemoresistance is one of the major challenges associated with this disease. It has been reported that epithelial-mesenchymal transition (EMT) is implicated in microRNA-driven modulation of tumor cells response to 5-fluorouracil and oxaliplatin. Moreover, from pharmacogenomic research, it is known that overexpression of genes encoding dihydropyrimidine dehydrogenase (*DPYD*), thymidylate synthase (*TYMS*), methylenetetrahydrofolate reductase (*MTHFR*), the DNA repair enzymes *ERCC1*, *ERCC2*, and *XRCC1*, and the phase 2 enzyme *GSTP1* impair the response to FOLFOX. It has been observed that EMT is associated with overexpression of *DPYD*, *TYMS*, *ERCC1*, and *GSTP1*. In this review, we investigated the role of miRNAs as EMT promotors in tumor cells, and its potential effect on the upregulation of *DPYD*, *TYMS*, *MTHFR*, *ERCC1*, *ERCC2*, *XRCC1,* and *GSTP1* expression, which would lead to resistance of CRC tumor cells to 5-fluorouracil and oxaliplatin. This constitutes a potential mechanism of epigenetic regulation involved in late-onset of acquired resistance in mCRC patients under FOLFOX chemotherapy. Expression of these biomarker microRNAs could serve as tools for personalized medicine, and as potential therapeutic targets in the future.

## 1. Introduction

Colorectal cancer (CRC) is defined as a malignant tumor that develops from the epithelial tissue of the colon or rectum, and is one of the most relevant malignant tumors worldwide [1,2]. The incidence and mortality rate of this disease has a direct association with the Human Developed Index (HDI), affecting developed, industrialized countries more severely than low HDI countries [3].

Treatment for CRC is defined based on the stage of the disease established by the TNM classification of the American Joint Committee on Cancer [4,5]. For patients diagnosed in stages I and II, the treatment usually implies resection of the tumor. Neoadjuvant or adjuvant chemotherapy regimens are indicated for patients diagnosed in stages III or IV, where there is an invasion of the lymphatic nodes or metastasis, and as palliative treatment for patients with non-resectable metastases [5].

Current schemes of chemotherapy for management of metastatic CRC (mCRC) are based on the use of 5-fluorouracil associated with leucovorin (5-FU/LV), or its prodrug capecitabine, in combination with a second cytotoxic agent such as oxaliplatin or irinotecan. 5-FU/LV in combination with oxaliplatin (L-OHP), also known as FOLFOX, is currently the most common first-line chemotherapeutic regimen, but other associations such as capecitabine plus L-OHP (CAPEOX), irinotecan plus 5-FU/LV (FOLFIRI), or combination of 5-FU/LV, L-OHP and irinotecan (FOLFOXIRI) are applied as well as alternative therapeutic schemes [5,6,7]. Biological therapies, such as EGFR or antiangiogenic inhibitors are more recently applied in association with conventional chemotherapy, in cases of advanced metastatic disease [5,7]. Nevertheless, these targeted therapies are not widely applied due to their prohibitive price and limited coverage of treatments from healthcare providers in most developing countries.

Although better screening, improved diagnostic techniques, and novel treatments have progressively increased the survival for patients diagnosed with CRC since 1970, patients diagnosed with distant-stage disease face unpromising odds with a five-year relative survival rate of only 14%, which is attributed to the development of acquired resistance to chemotherapy schemes [2,8]. For this reason, it is important to explore the molecular mechanisms involved in the development of chemoresistance in mCRC, to understand the pathways that contribute to the survival of tumor cells in patients undergoing chemotherapy, and to develop new therapeutic strategies that address these mechanisms.

## 2. EMT and Cancer Progression

Epithelial-mesenchymal transition (EMT) is a physiological process that implies a change of phenotype from an epithelial, polarized cell to a cell that exhibits mesenchymal characteristics, enabling cellular plasticity and adaptability, and is not limited to, but is often dysregulated in cancer cells. The reverse process, mesenchymal-epithelial transition (MET), may also occur in tumor cells. For example, EMT may allow the migration of cancer cells from the primary tumor to a metastatic focus, where new populations of cancer cells develop from the tumor propagating cells through undergoing MET. CSC may also develop from cancer cells through the activation of EMT. This population of cancer cells is characterized by a stem cell phenotype, low differentiation and proliferation rate, and inherent resistance to several chemotherapeutic agents [9,10,11].

A plethora of cancer cell populations, including mesenchymal, epithelial, hybrid, and CSCs, may coexist in the tumor tissue by proliferation and stochastic evolution of single precursor cancer cells. Disease progression, ischemia, chemotherapy, and many other factors may promote the diversification or selection of specific populations of cancer cells through mechanisms, including EMT and MET, and evidence has demonstrated that some of these populations are involved in the development of chemoresistance during disease progression [9,10].

β-Catenin is a protein that, in association with the transmembrane protein E-cadherin, forms part of the cell-cell interactions known as adherens junctions, which are characteristic structures of epithelial cells. Even though EMT can be triggered by several pathways, one of the pivotal steps of the EMT process is the upregulation of EMT-transcription factors (EMT-TFs) SNAIL, SLUG, ZEB, and TWIST, which repress E-cadherin expression. Free from the adherens junctions, cytosolic β-catenin may have two different fates. It is degraded by the destruction complex compounded by adenomatous polyposis coli (APC), a tumor suppressor frequently mutated in colorectal cancer, or either it translocates to the cell nucleus, where it interacts with T-cell factor (TCF)/Lymphoid enhancer-binding factor (Lef) family of transcription factors. This increases the expression of β-catenin target genes, which includes several oncogenes, and further upregulates the expression of EMT-TFs [12]. Tumor cells that acquire a mesenchymal phenotype through the execution of the EMT program gain traits that are relevant in the process of tumor progression and metastasis, including downregulation of the expression of cell adhesion molecules, detachment from surrounding cells, resistance to anoikis, and increased cell motility and invasiveness [12,13,14,15].

Because of the ongoing disease progression, the tumor cells plastically develop EMT, while EMT-TFs are dynamically and temporally expressed during this process, resulting in pleiotropic changes that affect the expression of several genes. Among others, proteins that influence FOLFOX response emerge as genes that may potentially be triggered by the EMT/MET dynamics during the evolution of the disease, contributing to the onset of acquired chemoresistance, which would add a new layer of complexity to the phenomena that lead to therapeutic failure in CRC.

## 3. Biomarkers of Response in Chemotherapy of Colorectal Cancer

Inherited genetic polymorphisms can affect the patient’s response to a certain drug, altering the outcomes of the treatment in terms of efficacy and safety. These variants often alter the function or expression of genes that have a role in pharmacokinetics or pharmacodynamics of the drug. When the scientific evidence supports the impact of a genetic polymorphism over the response of a specific drug or drug family, it is defined as a pharmacogenetic biomarker. One important example is the case of a single nucleotide variant of the thiopurine S-methyltransferase gene *TYMP**3A, related to severe adverse events in patients treated with thiopurines. Identification of this biomarker has meant an important improvement in the safety profile for patients treated with these drugs, and a huge success in the field of pharmacogenomics [16,17,18].

Pharmacogenetic biomarkers are important tools that serve to personalize treatments for a specific patient when the therapeutic window is narrow, and there is a high chance of severe adverse effects and therapeutic failure, which is frequent for cytotoxic chemotherapy schemes like FOLFOX. During the last years, pharmacogenomic studies have explored the relationship between genetic polymorphisms and the expression of certain genes that may explain the inter-individual variability in the response and toxicity profiles of CRC patients treated with FOLFOX chemotherapy. To date, several important associations have been identified.

### 3.1. 5-Pharmacogenomic Biomarkers of Fluorouracil Response

Briefly, 5-FU is an antimetabolite drug that exerts its cytotoxic effect mainly through inhibition of thymidylate synthase (TYMS) leading to dTMP depletion, impairing processes of DNA synthesis, and to a lesser extent through misincorporation of its metabolites into RNA and DNA [19,20] (Figure 1).

Polymorphic variants have been identified in the promoter enhancer region of the *TYMS* gene, consisting of a 5′UTR 28-bp double or triple tandem repeats (5′VNTR), and a 6bp indel in the 3′UTR (rs11280056 polymorphism). The triple tandem repeat (3R variant) and the insertion (ins) allele have been associated with increased *TYMS* expression and decreased survival in CRC patients treated with 5-FU based chemotherapy [5,21,22,23,24].

On the other hand, methylenetetrahydrofolate reductase (*MTHFR*) is an enzyme that catalyzes the conversion of 5-10-methylenetetrahydrofolate (5,10-methylene THF) into 5-methyltetrahydrofolate, which is a cofactor required for dTMP synthesis by *TYMS*. 5,10-Methylene THF stabilizes the binding of 5-FU metabolites to *TYMS* (Figure 1), which explains the improved response rate of 5-FU in association with leucovorin, compared with 5-FU as monotherapy [19]. Polymorphisms of the *MTHFR* gene have also been studied as potential biomarkers of 5-FU based chemotherapy response [20,21,22,24].

### 3.2. Pharmacogenomic Biomarkers of Oxaliplatin (L-OHP) Response

L-OHP is a platinum derivative that reacts with nucleotides in DNA strands, forming crosslinks that inhibit DNA synthesis and replication, leading to apoptosis of cancer cells (Figure 2). In this respect, DNA-repair protein complexes, responsible for repairing mismatches or abnormal nucleotides from DNA strands, often affect L-OHP efficacy and safety. The nucleotide excision repair pathway (NER), which involves multiple protein complexes that recognize, cleavage, and repair a fragment of the damaged DNA strand, is the main pathway involved in the repair of platinum-DNA adducts. Among the numerous subunits that constitute the NER pathway, the complex conformed by ERCC1/XPD proteins seems to be the limiting step defining the efficiency of the whole DNA repair process [8,24,25,26].

Polymorphisms in genes encoding *ERCC1* and *ERCC2* from the NER pathway have shown to have a role as biomarkers of both response and toxicity of FOLFOX regimens. The base excision repair (BER) pathway participates as well in the repair of platinum-DNA adducts, as polymorphisms of *XRCC1*, a critical subunit of the BER pathway, have shown to have an impact on clinical outcomes of patients undergoing FOLFOX chemotherapy [8,21,24,26,27].

On top of that, L-OHP and other platinum compounds are inactivated by glutathione-S transferases (GSTs). GSTs are a family of phase-2 enzymes that conjugate electrophilic xenobiotics to glutathione (such as platinum compounds) to facilitate their excretion. In this regard, metabolism of oxaliplatin is catalyzed mainly by the isoenzyme glutathione S-transferase π 1 (GSTP1), and polymorphisms of the *GSTP1* gene have been studied as biomarkers of both response and toxicity of FOLFOX regimens [8,21,24,26,27].

## 4. EMT and Expression of Biomarker Genes

Even though the evidence supports the role of genetic polymorphisms as predictors of chemotherapy response and toxicity, pharmacogenomic assessment of these variants is still not recommended in the routine clinical practice [5]. Genotyping studies often show inconsistencies regarding the effect of some genetic polymorphisms over relevant outcomes like overall survival, progression-free survival, or risk of adverse events, and sometimes the impact of these associations is marginal [5]. Therefore, the expression and function of *DPYD*, *TYMS*, *MTHFR*, *ERCC1*, *ERCC2*, *XRCC1,* and *GSTP1* may be influenced not only by genetic polymorphisms, but also by processes that are specific for the tumor tissue. As we describe in Table 1, it has been previously reported that the expression of the majority of FOLFOX biomarker genes is indeed affected during the EMT process.

Many studies have described the metabolic reprogramming of malignant cells during cancer progression. In this regard, some studies have revealed that *DPYD* expression is upregulated in cancer cell lines that present more mesenchymal characteristics [29]. Furthermore, it has been reported that knockdown of this gene in epithelial mammary cells during a TWIST-induced EMT program, and in hepatocellular carcinoma cell lines is enough to inhibit the transition process [28,29].

Following the same line, a different study showed that TYMS overexpression triggers EMT through upregulation of ZEB1 in NSCLC cell lines and vice versa, promoting 5-FU resistance and working as a sort of positive feedback between both genes where the specific mechanism remains to be explored [30]. Moreover, 5-FU resistant colon cancer cells overexpressing TYMS exhibited several mesenchymal traits and expression of molecular markers indicative of an ongoing EMT process, which points out that activity of the EMT program may have an impact on TYMS expression [31].

Upregulation of TWIST seems to increase the expression of both TYMS and DPYD, leading to 5-FU resistance in colon cancer cell lines [32]. The metabolic reprogramming may respond to increased nucleotide requirements in cancer cells undergoing EMT. Overall, these findings suggest that EMT-TFs may have a strong impact on chemosensitivity to 5-FU-based chemotherapy through upregulation of DPYD and TYMS expression.

Regarding genes that predict the response to oxaliplatin-based chemotherapy, upregulation of SNAIL, SLUG and ZEB1/ZEB2 have been linked to overexpression of ERCC1 in colon cancer, head and neck cancer, and NSCLC cell lines, which was associated with cisplatin resistance (another platinum derivative) in the latter [33,34,35]. GSTP1 also has shown to be upregulated in hepatocellular carcinoma cell lines by ZEB1 [37], suggesting that enzymes that confer resistance to L-OHP may be upregulated by EMT-TFs.

*XRCC1* seems to be the only gene that has been shown to be downregulated by EMT. Upregulation of the tumor suppressor circular RNA hsa_circ_0012563 increases expression of E-cadherin and XRCC1, downregulates N-cadherin expression, and diminishes invasion and migration in esophageal squamous carcinoma cell lines [36].

To summarize, the scientific evidence shows that several genes that impact the response to FOLFOX chemotherapy of CRC, including *DPYD*, *TYMS*, *ERCC1,* and *GSTP1* are upregulated as part of the EMT program, an association that has not been explored for *MTHFR* and *ERCC2*. Nevertheless, data collected from the Eukaryotic Promoter Database (http://epd.vital-it.ch) revealed that the human promoter regions of all of these genes, including the ones that remain unexplored, contain response elements to EMT-transcription factors, which include SLUG, ZEB1, TWIST1, β-catenin/TCF3, and β-catenin/TCF4, which further implies that these genes may be affected by the EMT process occurring during tumor progression (Figure 3).

As we previously mentioned, chemotherapy exerts an important selection pressure that may promote the survival of some cancer cell populations above others. The evidence has shown that cancer cells with active expression of EMT-TFs are more resistant to chemotherapeutic agents than cells with lower expression of these transcription factors [11]. The mechanism of chemoresistance in these cells may involve increased expression of genes that impair FOLFOX sensitivity at a cellular level. This phenomenon may constitute an underexplored mechanism of chemoresistance in tumor cells under EMT, implying that expression of the genes *DPYD*, *TYMS*, *MTHFR*, *ERCC1*, *ERCC2*, *XRCC1,* and *GSTP1* may be modulated EMT-TFs.

## 5. MiR Biogenesis and Cancer

Micro-RNAs (miRs) are a class or small sequences of non-coding RNA, about 19 to 24 nucleotides in length, that have been extensively studied during the last two decades for their role as epigenetic regulators of gene expression. These miRs are initially transcribed in the cell nucleus by RNA Polymerase II as a primary microRNA (pri-miRNA) which are later cleaved in the cell nucleus by the microprocessor-complex conformed by DROSHA and DGCR8. The resultant precursor microRNA (pre-miRNA) is exported to the cytoplasm where is further processed by the enzyme DICER1 into a small miR duplex, which is then unwound for one of the strands (guide strand) to be loaded into Argonaute, forming the microRNA-induced silencing complex (miRISC), while the other strand (passenger strand) is degraded [38,39].

The miRISC complex regulates gene expression by binding to the 3’ untranslated region of the target mRNA. If the miR guide strand and the target mRNA sequences are a perfect match, the mRNA is cleaved by the miRISC complex and degraded immediately. On the other hand, if the sequences are not perfectly aligned, the mRNA-miRISc complex is conveyed to the P-bodies for mRNA storage or decay. In both cases, translation of the target mRNA by the ribosome is generally downregulated by the influence of the miRISC-complex, and because of their short target sequence, one miR can target multiple mRNAs, exerting a profound effect on the cell phenotype [38,39].

In the past years, miRs have been described by their role in cancer progression, metastasis, and chemoresistance. For colorectal cancer, miR up or downregulation has been linked to critical pathways like inhibition of apoptosis, enhancement of DNA repair in tumor cells, upregulation of multidrug resistance membrane transporters, and regulation of cancer stem cell (CSC) population in the tumor tissue [38,39,40,41,42]. MiR expression has been associated with response to chemotherapy in clinical trials and resistance to platinum derivatives, fluoropyrimidines, and other chemotherapeutic agents [43]. Nonetheless, the association between known pharmacogenomic biomarkers of response to CRC chemotherapy and miR expression has not been deeply explored.

### 5.1. MiRs as Enhancers of FOLFOX Chemosensitivity

Several miRs have been shown to play an important role in improving 5-FU and L-OHP sensitivity in colon cancer cell lines. This improvement in chemotherapy response has been associated with the modulation of pathways usually dysregulated in cancer, as described in Table 2. These include the targeting of transcription factors specificity protein 1 (Sp1) by miR-125b-5p [44], nuclear factor kappa B subunit 1 (p50) by miR-15 [45], ETS proto-oncogene 1, transcription factor (ETS1) by miR-532-3p, and forkhead box M1 (FOXM1) by miR-149 [46], and miR-320 [47]. These transcription factors participate in promoting the progression of the cell cycle and proliferation and often trigger EMT in cancer cells.

Histone methyltransferases are important regulators of gene expression that make DNA more or less accessible to RNA polymerases. MiR-133b targets DOT1L, a histone H3 lysine-79 specific N-methyltransferase upregulated in colon cancer cells-derived spheroids, reducing CSC phenotype, and improving sensitivity to 5-FU and L-OHP [48]. High mobility group protein A2 (HMGA2) is a non-histone chromosomal protein that also contributes to transcription regulation through the regulation of chromosomal condensation. Downregulation of HMGA2 expression by miR-204 improves in vitro 5-FU-chemosensitivity by inhibiting the PI3K/AKT pathway [57].

Targeting anti-apoptotic proteins has also been described as a mechanism that improves the sensitivity of cancer cells to chemotherapeutics. X-linked inhibitor of apoptosis (XIAP) is targeted by miR-122 [52] and miR-874 [59], and B-cell lymphoma 2 apoptosis regulator (BCL2) is targeted by miR-139-5p [56]. MiR-15 has shown to indirectly downregulate BCL2 and B-cell lymphoma-extra large apoptosis regulator (BCL-XL) via downregulation of NF-κB signaling [45]. Overall, the downregulation of these anti-apoptotic proteins improves apoptosis induced by 5-FU and L-OHP. MiR-224-5p was downregulated in 5-FU-resistant cell lines, and although no specific targets were specified, transient overexpression of miR-224-5p led to increased caspase 3/7 activation in these 5-FU-resistant cells lines after 5-FU treatment [50]. It is noticeable that downregulation of BCL2 by miR-139-5p correlates with inhibition of EMT, suggesting that the transition is part of the mechanisms involved in improving chemosensitivity, and the possibility that miR-139-5p could potentially target other mRNAs that regulate EMT [45].

Tumor metabolic reprogramming is another mechanism that is targeted by miRs. Upregulation of glycerophosphodiester phosphodiesterase domain containing 5 (GDPD5, an enzyme that participates in glycerol metabolism), superoxide dismutase 2 (SOD2, involved in superoxide radicals detoxification), and heat shock protein beta-1 (Hsp27, a chaperone involved in cellular stress-response), occurs in several CRC samples and colon cancer cell lines. MiR-195-5p [53], miR-324-5p [54], and miR-214 [49] target GDPD5, SOD2, and Hsp27, respectively. Moreover, downregulation of GDPD5 and SOD2 by these miRs or by small interfering RNAs also reverse EMT in colon cancer cell lines in vitro [53,54].

MiR-330 and miR494 are the only described miRs that target an enzyme directly associated with 5-FU response. Further, 5-FU-resistant cells overexpress *DPYD* acquiring resistance to 5-FU, which was reversed with transfection with miR-494 mimics, and miR-330 increased 5-FU-chemosensitivity in colon cancer cell lines through direct downregulation of *TYMS* expression, both increasing apoptosis induced by this chemotherapeutic agent in colon cancer cells [55,58].

### 5.2. MiRs Promoting FOLFOX Chemoresistance

On the contrary, evidence shows that miRs also contribute to FOLFOX chemoresistance and are upregulated in many induced 5-FU or L-OHP resistant cell lines, as presented in Table 3. Tumor suppressors targeted by miRs include phosphatase and tensin homolog (PTEN), targeted by miR-543 [60] and miR-21 [61], cyclin-dependent kinase inhibitor 1A (p21) targeted by miR-520g [62] and indirectly by miR-543, APC targeted by miR-125p, glycogen synthase kinase-3 beta (Gsk3β) targeted by miR-199a/b, F-box and WD repeat domain containing 7 (FBXW7, a subunit of the SCF ubiquitin ligase complex that participates in the degradation of several proteins involved in mitogenic pathways), targeted by miR-92a-3p [63], and Programmed Cell Death 4 (PDCD4) which is targeted by miR-21 [61].

Regarding pro-apoptotic proteins targeted by miRs, previous research has shown that BCL2 interacting protein 2 (BNIP2) is targeted by miR-20a [68], the modulator of apoptosis 1 (MOAP1) is targeted by miR-92a-3p [63], and BCL2 associated X protein (BAX) which is indirectly downregulated by miR-543 [60] downregulating the expression of these pro-apoptotic mediators. Induced overexpression of miR-625-3p directly downregulates the expression of mitogen-activated protein kinase 6 (MAP2K6), inhibiting phosphorylation of the MAP kinase p38 alpha (p38), downstream p38-mediated signaling, and control of the cell cycle, ultimately impairing L-OHP-mediated apoptosis [67]. Downregulation of these proteins by miRs inhibits 5-FU and L-OHP induced cytotoxicity and promotes cell survival. MiR-543 not only has shown not only to target and downregulate PTEN, but its overexpression also correlates with upregulation of BCL2 and activation of AKT, further inhibiting in-vitro apoptosis [60].

MiR-125b-5p seems to function both as a tumor suppressor miR in induced L-OHP-resistant HCT8 cells, or conversely as an oncomiR (oncogenic miR) when it is upregulated in HCT116 and SW620 cell lines in response to treatment with C-X-C motif chemokine ligand 12 (CXCL12), leading to chemoresistance. This phenomenon may account for the specific responses that cancer cells exhibit under different stimuli [44,64].

A common finding among the cited works is that miRs trigger EMT as part of the mechanism involved in the acquisition of resistance to 5-FU and L-OHP in cancer cell lines, which is associated with expression of miR-21, miR-92a-3p, miR-125b-5p, miR-199a/b, and miR-210. Inhibition of these miRs leads to the restoration of an epithelial phenotype and restoration of chemosensitivity to 5-FU and L-OHP [61,63,66]. Acquisition of an intermediate epithelial-mesenchymal phenotype from an incomplete EMT is associated with miR-23b overexpression in L-OHP-resistant colon cancer cells, and knockdown of this miR leads to the completion of the EMT process and acquisition of a mesenchymal phenotype, restoring the sensitivity to L-OHP [50].

Moreover, inhibition of EMT is also a common feature observed for miRs that sensitize resistant colon cancer cell lines to 5-FU and L-OHP. As we previously mentioned, upregulation of miR-125b-5p, miR-195-5p, miR-139-5p, miR-324-5p, miR-320, and miR-532-5p have shown to inhibit the EMT process in colon cancer cells, leading to the acquisition of an epithelial phenotype, and restoring sensitivity to 5-FU and L-OHP in vitro [44,51,53,54,56]. Downregulation of FOXM1 by miRs also leads to inhibition of EMT and suppression of invasive phenotype in colon cancer cell lines, combined with restoration of chemosensitivity [47,69].

The role of several miRs in promoting FOLFOX chemoresistance has also been reported for non-in vitro settings. Analysis in a cohort of CRC patients found that high expression of mir-21 was associated with poor therapeutic outcome in patients receiving 5-FU-based chemotherapy [70]. MiR-19a and miR-17-5p have been proposed as potential biomarkers of FOLFOX chemoresistance in CRC patients, and expression of miR-27b, miR-181b, and miR-625-3p have been associated with poor response to FOLFOX and CAPEOX [71]. In animal models, miR-92a-3p expression promoted resistance to 5-FU/L-OHP therapy, and its expression was directly correlated with poor survival and chemoresistance in a cohort of CRC patients [63]. This confirms that the effect of miRs in promoting chemoresistance is not limited to in vitro models, and their impact on chemotherapy response to FOLFOX needs to be further explored.

To summarize, the impact of miRs on sensitivity to 5-FU and L-OHP has been attributed to several targets, including modulation of oncogene/tumor suppressor expression, regulation of proapoptotic/antiapoptotic proteins, metabolic reprogramming, and acquisition of CSC phenotype. Nevertheless, except for miR-494 that targets *DPYD* mRNA and miR-330 that targets *TYMS* mRNA, none of these studies assess the impact of miRs over the described FOLFOX biomarkers that directly interact with these drugs, which may be downstream modulated by EMT-TFs transcription factors or other pathways, contributing to the acquisition of chemoresistance. Although the research in the field is scarce, the evidence we present in this review suggests that microRNAs may potentially affect chemotherapy response by modulating the expression of *DPYD*, *TYMS*, *MTHFR*, *ERCC1*, *ERCC2*, *XRCC1,* and *GSTP1,* genes currently known to influence chemotherapy response. We propose microRNAs as new promising biomarkers for chemoresistance development in the context of disease progression, opening a new field of epigenetic modulation in pharmacogenomics.

## 6. Concluding Remarks and Future Perspectives

Over the last few years, it has been assumed that tumor cells that exhibit a mesenchymal phenotype are inherently more resistant to chemotherapy compared to tumor cells with a well-differentiated epithelial phenotype. This phenomenon has been attributed to their reduced proliferative activity and increased expression of efflux transporters, such as the ATP-binding cassette family proteins, but the relationship between the targets that interact directly with cytotoxic drugs in these phenotypically altered conditions is lesser-known.

In this review, we have collected evidence that suggests that miRs have a critical role in promoting FOLFOX chemoresistance in CRC. Indeed, several studies have shown that miRs modulate 5-FU and L-OHP sensitivity in vitro and that, in many cases, the chemoresistance involved or was correlated with EMT and expression of mesenchymal markers.

Cancer cell plasticity is one of the characteristics that makes cancer treatment such a difficult task, and EMT-MET are both pivotal mechanisms that allow cancer cells to eventually develop resistance to the therapeutic strategies currently available [9,10]. It is important to remark that the transition is not an immediate process and that generally cancer cells do not go from absolute epithelial to absolute mesenchymal in a short period. EMT and MET in cancer cells occur dynamically, and cancer cells in the tumor tissue often exhibit different degrees of hybrid epithelial-mesenchymal phenotype, as part of the tumor heterogeneity. For example, miR-23b seems to promote this intermediate phenotype in HCT116 cells. Knockout of this miR caused acquisition of a mesenchymal phenotype of colon cancer cells in vitro, which interestingly, restored L-OHP sensitivity [50]. This evidence suggests that a hybrid epithelial-mesenchymal phenotype may be more resistant to 5-FU and L-OHP. Non-cancer cells in the tumor tissue may also show some degree of transition, including cancer-associated fibroblasts that may also contribute to the diversity of miRs [63].

Herein, we have also presented evidence that supports the hypothesis that FOLFOX biomarker genes may be regulated by EMT transcription factors. Upregulation of *TYMS* and *DPYD* seems to be a requirement for EMT in certain types of experimental models [29,30]. *DPYD* and *TYMS* are genes that are involved in pyrimidine biosynthesis, *MTHFR* is involved in folate metabolism, and *GSTP1* is an enzyme involved in xenobiotic detoxification. Many of these pathways may be dysregulated in cancer cells to cope with increased proliferative activity of the tumor tissue. In the same way, *ERCC1* and *ERCC2* from the NER pathway may respond to avoid replicative stress as a consequence of exacerbated DNA replication. *XRCC1* from the BER pathway seems to be an exception, and EMT-TFs may function as repressors of its expression in cancer tissues in the same way that E-cadherin expression is regulated by these transcription factors, although more research is needed on this topic. The change in expression of FOLFOX biomarker genes may constitute a metabolic reprogramming that functions as an adaptative mechanism of tumor cells during cancer progression. In this regard, EMT may be the process triggering this metabolic reprogramming through EMT-TFs, as part of disease progression or triggered by external stimuli, such as chemotherapy, allowing these hypothetical hybrid epithelial-mesenchymal tumor cells to tolerate FOLFOX chemotherapy. MiRs would function as an epigenetic regulator that may participate in late-onset chemotherapy resistance in mCRC patients undergoing FOLFOX chemotherapy by triggering EMT (Figure 4).

Genetic polymorphisms may alter a protein when they change an important amino acid required for its proper function. On the flipside, polymorphisms may not alter the gene coding sequence, but affect mRNA processivity (referring to the ability of the RNA Pol II to achieve complete mRNA elongation of the gene without disassembling prematurely), stability or alter miR targeting sites, modifying the protein expression. These are germline variants that permanently affect protein expression or function during an individual lifetime. Genetic polymorphisms of *DPYD*, *TYMS*, *MTHFR*, *ERCC1*, *ERCC2*, *XRCC1,* and *GSTP1* have been shown to influence FOLFOX response in mCRC patients [21]. Nevertheless, these polymorphisms do not explain changes in the expression of these genes that may occur during the disease progression as consequence of cancer cell plasticity.

Expression of miRs that trigger EMT may potentiate the effect of these polymorphisms resulting in a high chance of therapeutic failure. For example, the *TYMS* in allele (rs11280056) is associated with higher *TYMS* expression and lower OS in stage IV CRC patients. High expression of miR-125b-5p or miR-199a/b is associated with increased active Wnt/β-catenin and expression of β-catenin target genes including ZEB1 [64,65], which has shown to trigger TYMS expression [30]. Knowing these findings, we could expect a high chance of resistance to 5-FU in a patient that presents high levels of miR-125b-5p or miR-199a/b expression in combination with the *TYMS* ins allele. These associations need to be validated by experimental evidence and clinical research in the future. Nonetheless, previous research suggests that intratumoral levels of miRs are directly correlated with levels of this miR found in exosomes isolated from plasma in CRC patients [63,72]. This would make miRs an excellent tool to predict FOLFOX response and development of resistance through non-invasive management techniques in combination with known biomarker polymorphisms.

Currently, assessment of biomarkers in CRC is mostly indicated when specific treatments are prescribed, such as *RAS* mutation status in patients before treatment with EGFR-targeted monoclonal antibodies. If the hypothesis proposed in this review is confirmed by scientific evidence in the future, specific miRs may account as indicators of cancer plasticity in the tumor foci in the same way that genetic polymorphisms currently account for interindividual variability. Combination of biomarker miRs and genetic polymorphisms could represent better tools not only to predict FOLFOX response, but also to track the evolution of FOLFOX sensitivity during the disease progression. Establishing pharmacogenomic biomarker miRs for chemotherapy response would mean that acquired resistance could be assessed through a blood sample in the routine clinical practice in the future.

Finally, the potential of miRs as a direct therapeutic target for novel treatment development seems like a promising alternative for patients who are non-responsive to first-line chemotherapy schemes due to intrinsic or acquired chemoresistance. As we summarized in Table 1, miRs have the potential to sensitize colon cancer cell lines to 5-FU and L-OHP through inhibition of EMT, downregulation of anti-apoptotic proteins, and by directly targeting FOLFOX biomarker mRNAs. Delivery of miR-mimics for these miRs or antagomiRs that antagonize chemoresistance-promoting miRs may be a promising therapeutic strategy to exploit the potential of miRs to improve FOLFOX response. The development of optimal tools for delivery of miRs through extracellular vesicles to the tumor tissue seems to be the major challenge in the development of miR-based treatments, but to date, several strategies are being tested in clinical trials [73].

In conclusion, we propose a potential role of miRs as promotors of acquired chemoresistance to FOLFOX chemotherapy, by upregulating the expression of *DPYD*, *TYMS*, *MTHFR*, *ERCC1*, *ERCC2*, *XRCC1*, and *GSTP1* in cancer cells via stimulating EMT-TFs. This potential novel mechanism of epigenetic-induced chemoresistance needs to be explored to open new possibilities of improved chemotherapy response and prognostic in patients receiving FOLFOX chemotherapy.

## Figures and Tables

**Figure 1 pharmaceutics-13-00075-f001:**
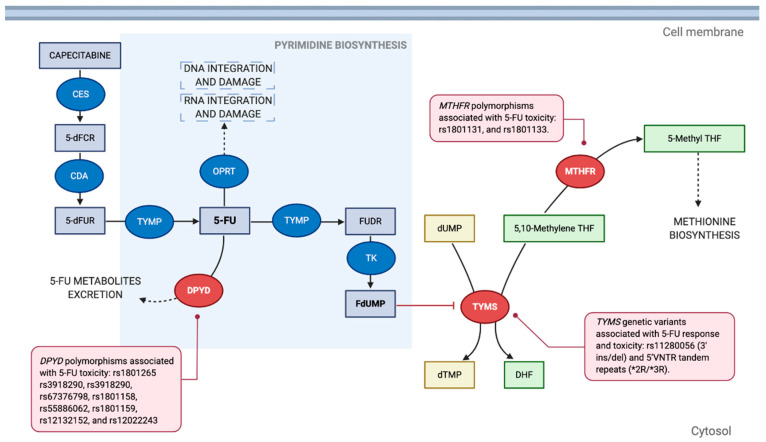
**Mechanism of action of 5-fluorouracil:** Capecitabine is a prodrug that is converted to 5-FU through metabolization by the enzymes carboxylesterase (CES) into 5′-deoxy-5-fluorocytidine (5′-dFCR), then by cytidine deaminase (CDA) into doxifluridine (5′-dFUR), and finally by thymidine phosphorylase (TYMP) into 5-FU. 5-FU is metabolized by TYMP into floxuridine (FUDR), and then by thymidine kinase (TK) into 5-fluoro-2′-deoxyuridine 5′-monophosphate (FdUMP), which inhibits its main therapeutic target thymidylate synthase (TYMS) by competing with its natural ligand deoxyuridine monophosphate (dUMP). TYMS normally transfers methyl groups from 5-10-methylenetetrahydrofolate (5,10-methylene THF) into dUMP to obtain deoxythymidine monophosphate (dTMP) and dihydrofolate (DHF), which is the rate-limiting step in the synthesis of deoxythymidine nucleotides required for DNA replication. 5-FU can be directly inactivated by dihydropyrimidine dehydrogenase (DPYD) into metabolites that are eliminated. Binding of FdUMP to TYMS is stabilized by 5,10-methylene THF, and metabolization of this cofactor into 5-methyltetrahydrofolate (5-methyl THF) by methylenetetrahydrofolate reductase (MTHFR) reduces FdUMP affinity to TYMS. 5-FU can also be metabolized by orotate phosphoribosyltransferase (OPRT) into 5-FU metabolites that are incorporated into the RNA and DNA of the cell, accounting to a lesser extent for 5-FU cytotoxicity. Polymorphisms of *DPYD*, *TYMS*, and *MTHFR* coding genes have been associated with 5-FU response [21]. Enzymes are shown as ovals and metabolites as rectangles. Adapted from Longley et al. [19] and Ulrich et al. [20]. Created with BioRender.com.

**Figure 2 pharmaceutics-13-00075-f002:**
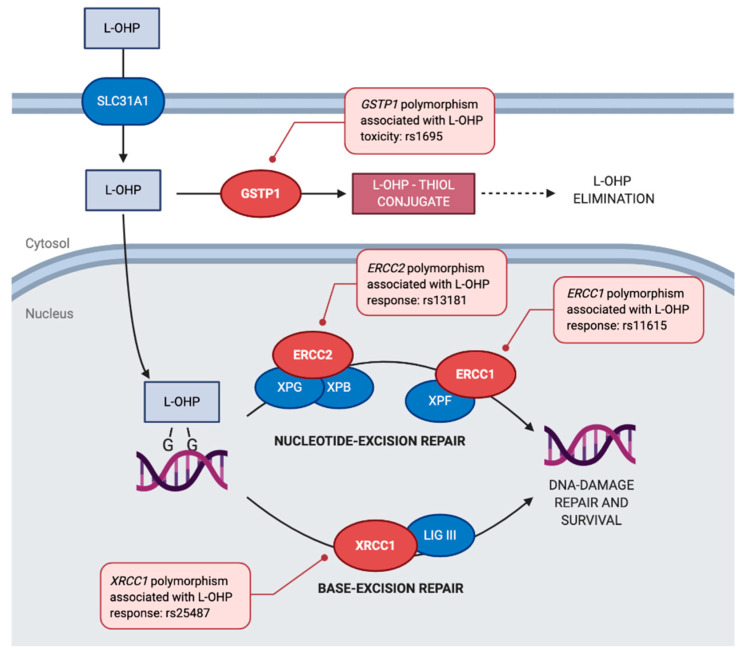
**Mechanism of action of Oxaliplatin:** L-OHP is transported into the cell by solute carrier family 31 (copper transporter), member 1 (SLC31A1). Inside the cell, L-OHP forms adducts with the DNA, in particular with guanine bases, forming DNA crosslinks that induce apoptosis of the cell. Platinum-DNA adducts can be repaired by the nucleotide-excision repair pathway (NER), which involves several steps and enzyme complexes, including xeroderma pigmentosum, complementation group G (XPG) which interacts with the TFIIH core complex helicase, composed by the excision repair cross-complementation group 2 (ERCC2) and xeroderma pigmentosum group B-complementing protein (XPB), unwinding the damaged DNA strand. This step is followed by an excision step of the damaged DNA fragment catalyzed by the endonuclease complex formed by the excision repair cross-complementation group 1 (ERCC1) and xeroderma pigmentosum, complementation group F (XPF), which is followed by the synthesis of a new DNA strand. Alternatively, platinum-DNA adducts can be repaired by the base-excision repair pathway (BER), in which the last step involves the binding of DNA 5′ and 3′ ends by the participation of X-ray repair cross-complementing 1 (XRCC1) and DNA ligase III (Lig III). Both pathways lead to the elimination of platinum-DNA adducts inhibiting DNA damage driven apoptosis. L-OHP can be directly detoxified by glutathione S-transferase π 1 (GSTP1) into L-OHP-thiol conjugates that are eliminated. Polymorphisms of *ERCC1*, *ERCC2*, *XRCC1*, and *GSTP1* coding genes have been associated with L-OHP response [21]. Several enzyme complexes from the NER and BER pathways are omitted for simplicity. Enzymes are shown as ovals and metabolites as rectangles. Adapted from Marsh et al. [25]. Created with BioRender.com.

**Figure 3 pharmaceutics-13-00075-f003:**
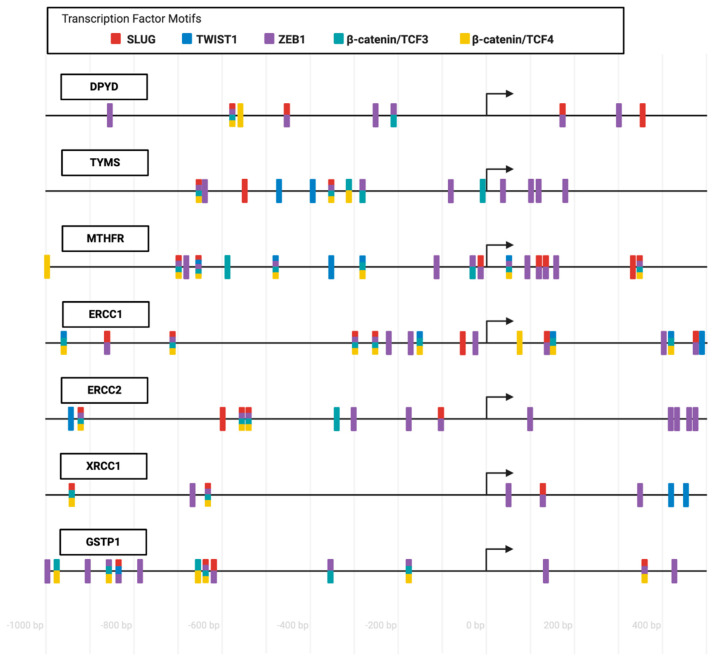
**FOLFOX biomarker genes promoters contain several EMT-TFs response elements:** The promoter regions of the genes *DPYD*, *TYMS*, *MTHFR*, *ERCC1*, *ERCC2*, *XRCC1*, and *GSTP1* are represented from 1000 bp upstream to 500 bp downstream to the transcription start site (represented by arrows). Transcription factor motifs are represented in red rectangles for SLUG, in blue rectangles for TWIST1, in purple rectangles for ZEB1, in turquoise rectangles for β-catenin/TCF3, and in yellow rectangles for β-catenin/TCF4. More than one color in a position indicates that a transcription factor motif is common for more than one of the represented transcription factors. The transcription factor motifs positions are approximated and were obtained from the Eukaryotic Promoter Database (http://epd.vital-it.ch). Created with BioRender.com.

**Figure 4 pharmaceutics-13-00075-f004:**
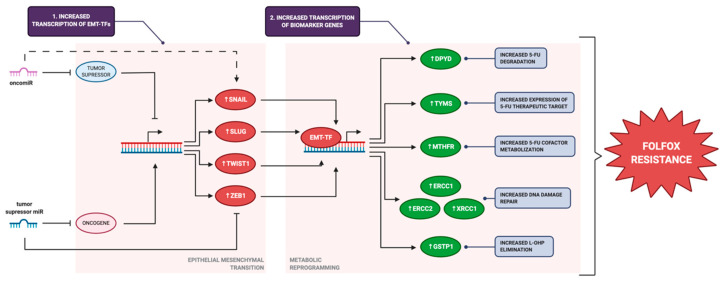
**Proposed mechanism for miRs participation in response to FOLFOX chemotherapy:** 1.-OncomiRs upregulate transcription of epithelial-mesenchymal transition-transcription factors (EMT-TFs) by downregulation of expression of tumor suppressors, or directly upregulating their expression by mechanisms that remain to be explored, triggering epithelial-mesenchymal transition (EMT) in cancer cells. Tumor suppressor miRs downregulate the expression of EMT-TFs by downregulation of oncogenes that upregulate their transcription, or by directly targeting EMT-TFs mRNA. 2.-EMT-TFs SNAIL, SLUG, TWIST1, ZEB1 directly upregulate transcription of FOLFOX biomarker genes as part of metabolic reprogramming, allowing adaptation of the cancer cell to increased proliferation and replication stress. These metabolic changes triggered by progressive EMT in cancer cells affect FOLFOX response by increasing 5-FU and L-OHP elimination (DPYD, GSTP1), increasing expression of 5-FU therapeutic target (TYMS), decreasing levels of 5,10-methylene THF required for proper TYMS inhibition (MTHFR), and increasing DNA damage repair and DNA damage-driven apoptosis (ERCC1, ERCC2, XRCC1). Overall, this would lead to FOLFOX resistance.

**Table 1 pharmaceutics-13-00075-t001:** Expression of FOLFOX biomarker genes is modified by EMT.

Biomarker Genes	Effect of EMT in Expression	Mechanism of Regulation	Reference
**DPYD**	Upregulation	Direct correlation between expression of mesenchymal markers and DPYD expression in hepatocellular carcinoma samples. DPYD facilitates EMT by suppressing the expression of p38 in HCCLM3 cell lines.	Zhu et al. [28]
**DPYD**	Upregulation	Direct correlation between mesenchymal phenotype and DPYD expression in cancer cell lines. DPYD knockdown suppresses Twist-induced EMT in mammospheres derived from HMEC cells.	Shaul et al. [29]
**TYMS**	Upregulation	TYMS is upregulated in mesenchymal-like compared to epithelial-like cancer cells. ZEB1 upregulates TYMS through miR-375 downregulation. TYMS overexpression upregulates ZEB1 in A549 cells.	Sidiqqui et al. [30]
**TYMS**	Upregulation	5-FU-resistant HCT116 cells increased TYMS expression via HSP90/Src, which correlates with downregulation of E-cadherin.	Ahn et al. [31]
**DPYD, TYMS**	Upregulation	TWIST silencing downregulates DPYD and TYMS and improves 5-FU sensitivity in HT29 cells.	Sakowicz-Burkiewicz et al. [32]
**ERCC1**	Upregulation	L-OHP-resistant HCT116 cells upregulate ERCC1 via the AKT pathway and SLUG expression.	Wei et al. [33]
**ERCC1**	Upregulation	Cisplatin-resistant A549 cells upregulate ERCC1 via ZEB1/2.	Wu et al. [34]
**ERCC1**	Upregulation	Direct correlation between SNAIL and ERCC1 expression in cancer cell lines. Overexpression of SNAIL upregulates ERCC1 in FaDu and CAL-27 cell lines.	Hsu et al. [35]
**XRCC1**	Downregulation	Downregulation of hsa_circ_0012563 in TE-1 cell lines upregulates XRCC1 and E-cadherin.	Zhang et al. [36]
**GSTP1**	Upregulation	ZEB1 promotes paclitaxel resistance through upregulation of GSTP1 in Huh7 and HCCLM3 cell lines.	Yang et al. [37]

**Table 2 pharmaceutics-13-00075-t002:** miRs improving sensitivity to 5-FU and L-OHP in colon cancer cell lines.

miR	Effect on Chemotherapy	Target Genes	Mechanism of Sensitization	Reference
**miR-125b-5p**	Downregulated in 5-FU, and L-OHP-resistant colon cancer cell lines.	Sp1	Targets Sp1 inhibiting overexpression of CD248 and EMT in 5-FU and L-OHP resistant HCT8 cells.	Park et al. [44]
**miR-133b**	Sensitizes colon cancer cell lines to 5-FU, and L-OHP.	DOT1L	Downregulates DOT1L promoting differentiation and reducing stemness in spheroids derived from HT29 and SW480 cells.	Lv et al. [48]
**miR-214**	Sensitizes colon cancer cell lines to 5-FU.	Hsp27	Downregulates Hsp27 and enhances caspase-3 activation in 5-FU-treated HT29 and LoVo cells.	Yang et al. [49]
**miR-224-5p**	Sensitizes colon cancer cell lines to 5-FU.	Not specified	Downregulated in 5-FU resistant HCT116 and DLD-1 cells. Induced overexpression restores sensitivity to 5-FU increasing apoptosis in HCT116 cells.	Gasiulė et al. [50]
**miR-532-3p**	Sensitizes colon cancer cell lines to 5-FU and cisplatin	ETS1, TGM2	Inhibits Wnt/β-catenin signaling mediated by ETS1 and TGM2 expression in HT29 and RKO cells.	Gu et al. [51]
**miR-15**	Sensitizes colon cancer cell lines to 5-FU, and L-OHP.	p50	Downregulates expression of p50 subunit of NF-κB inhibiting upregulation of antiapoptotic factors BCL-2 and BCL-XL in HCT116 cells.	Liu et al. [45]
**miR-122**	Sensitizes colon cancer cell lines to L-OHP.	XIAP	Enhances apoptosis and chemosensitivity in L-OHP-resistant HCT116 and SW620 cells.	Hua et al. [52]
**miR-195-5p**	Sensitizes colon cancer cell lines to 5-FU, and L-OHP.	GDPD5	5-FU-resistant HCT116 cells transfected with miR-195-5p restore sensitivity to 5-FU.	Feng et al. [53]
**miR-324-5p**	Induced overexpression sensitizes colon cancer cell lines to 5-FU/L-OHP.	SOD2	Downregulates SOD2 aberrant overexpression, expression of EMT markers, and CSC phenotype in HCT116 and DLD-1 cells, restoring sensitivity to FOLFOX-like treatment.	Bamodu et al. [54]
**miR-330**	Sensitizes colon cancer cell lines to 5-FU.	TYMS	Downregulates TYMS expression and increases 5-FU-mediated apoptosis in HCT116 and HT29 cells.	Xu et al. [55]
**miR-139-5p**	Sensitizes colon cancer cell lines to 5-FU, and L-OHP.	BCL-2	Enhances apoptosis and inhibits EMT promoting differentiation of HCT116 and SW620 cells.	Li et al. [56]
**miR-149**	Sensitizes colon cancer cell lines to 5-FU.	FOXM1	Downregulates expression of FOXM1 in 5-FU-resistant HCT8 and LoVo cell lines.	Liu et al. [46]
**miR-204**	Sensitizes colon cancer cell lines to 5-FU.	HMGA2	Downregulates expression of HMGA2 in HCT116 and SW480 cell lines, downstream inhibiting PI3K/AKT pathway activation.	Wu et al. [57]
**miR-874**	Sensitizes colon cancer cell lines to 5-FU.	XIAP	Inhibits proliferation, colony formation and enhances apoptosis and 5-FU sensitivity in SW480 cells.	Han et al. [53]
**miR-320**	Sensitizes colon cancer cell lines to 5-FU, and L-OHP.	FOXM1	Downregulates expression of FOXM1, and downstream expression of β-catenin, reducing viability, migration and invasion of HCT116 and HT29 cells.	Wan et al. [47]
**miR-494**	Sensitizes colon cancer cell lines to 5-FU.	DPYD	Downregulates DPYD expression and increases 5-FU bioavailability in 5-FU resistant SW480 cells.	Chai et al. [58]

**Table 3 pharmaceutics-13-00075-t003:** miRs impairing sensitivity to 5-FU and L-OHP in colon cancer cell lines.

miR	Effect on Chemotherapy	Target Genes	Mechanism of Resistance	Reference
**miR-92a-3p**	Reduces sensitivity to 5-FU/L-OHP in colon cancer cell lines.	FBXW7, MOAP1	Induces EMT by inhibition of β-catenin degradation and inhibits apoptosis induced by FOLFOX-like treatment in SW480, SW620, and LOVO cell lines.	Hu et al. [63]
**miR-23b**	Upregulated in L-OHP-resistant colon cancer cell lines.	Several	Upregulated in L-OHP-resistant HCT116 cells, which exhibit a hybrid epithelial-mesenchymal phenotype. Knock out or miR-23b restores L-OHP sensitivity and induces acquisition of a defined mesenchymal phenotype in HCT116 cells.	Gasiulė et al. [50]
**miR-543**	Reduces sensitivity to 5-FU in colon cancer cell lines.	PTEN	Promotes proliferation of HCT8 cells and increases IC50 of 5-FU. Downregulates expression of PTEN, p53, p21, and BAX.	Liu et al. [60]
**miR-125b-5p**	Reduces sensitivity to 5-FU in colon cancer cell lines.	APC	Downregulates APC promoting Wnt/β-catenin pathway signaling and autophagy in HCT116 and SW620 cells.	Yu et al. [64]
**miR-199a/b**	Reduces sensitivity to cisplatin in colon cancer cell lines.	Gsk3β	Upregulated in patient-derived colon cancer stem cells, downregulates Gsk3β promoting Wnt/β-catenin pathway signaling.	Chen et al. [65]
**miR-210**	Reduces sensitivity to 5-FU/L-OHP in colon cancer cell lines.	Not specified	Associated with EMT in HCT-8 cells, leading to resistance to anoikis and reduced response to FOLFOX-like treatment.	Bigagli et al. [66]
**miR-520g**	Reduces sensitivity to 5-FU, and L-OHP in colon cancer cell lines.	p21	Downregulates expression of p21 inhibiting p53-mediated apoptosis in HCT116 and RKO cells.	Zhang et al. [62]
**miR-625-3p**	Reduces sensitivity to L-OHP in colon cancer cell lines.	MAP2K6	Inhibits the MAP2K6/p38-pathway and p38-mediated apoptosis and cell cycle control in HCT116 and SW620 cells.	Rasmussen et al. [67]
**miR-21**	Reduces sensitivity to 5-FU/L-OHP in colon cancer cell lines.	PTEN, PDCD4, TGFβ-R2	Activates β-catenin/TCF-mediated gene transcription, leading to EMT in HCT116 cell lines.	Yu et al. [61]
**miR-20a**	Reduces sensitivity to 5-FU, and L-OHP in colon cancer cell lines.	BNIP2	Inhibits 5-FU and L-OHP mediated apoptosis through downregulation of proapoptotic protein BNIP2 in SW480 and SW620 cells.	Chai et al. [68]

## Data Availability

Data sharing is not applicable to this article.

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
