# Peer review of "Epithelial-Mesenchymal Transition and MicroRNAs in Colorectal Cancer Chemoresistance to FOLFOX"

_pharmaceutics, 2021, doi:10.3390/pharmaceutics13010075_

Round 1

Reviewer 1 Report

The authors (Escalante et al) here present a review investigating the role of miRNAs as modifiers of FOLFOX chemotherapy resistance and claim to show this resistance comes about via modulation of genes involved in epithelial to mesenchymal transition (EMT). While the review is well written and thorough, it does tread similar ground to other reviews on the topic. Several points raise concern which will require revision of the manuscript.

  • The tile would lead a reader to suggest this paper will focus heavily on miRNAs however the first 50% of the paper deals only with EMT. It would be of my opinion that the miRNA information should be better integrated into the manuscript or the title should be modified in order to better reflect the data within the paper
  • Title: Misspelling in title: trough - through
  • Line 57: The use of targeted therapies: The use of targeted therapies is now predominantly first line: cetuximab is used as a first line agent in patents with Kras wild type and bevacizumab is given in combo with folfox/ folfiri in Kras mutant CRC.
  • Line 120: The title of the section is unclear and needs to be reworded
  • Line 152-53: Reference needed regarding the improvement of therapeutic response with the addition of folinic acid to 5-FU 
  • Line 156: The title of the section is unclear and needs to be reworded
  • Line 165: Reference needed
  • Pay attention to table numbering. Two table 1 exist.
  • Line 201-206 This statement need to be backed up with references.
  • Section 5.1 and 5.2 miRNAs: A discussion here about the clinical relevance of these markers would be appropriate. Most if not all of these studies have been performed in vitro in well established cell lines. It is worth noting that the cell lines are generated to develop resistance rather that multi-clonal populations that exist in a tumour.  Have these miRNAs been shown to contribute to clinical acquired resistance or clinical response? This would add value to the review and interest to the reader. Even an animal study delivering antagomirs would show relevance. 
  • A clear link between EMT and the the apoptosis regulators is not clear. The section is difficult to follow and it is not certain if the Mir targeting the apoptotic modulators is responsible for the EMT phenotype as well.

Author Response

We greatly appreciate the pertinent observations of the referee, which have improved considerably our manuscript. Below you will see the point-by-point response.
  1. The tile would lead a reader to suggest this paper will focus heavily on miRNAs however the first 50% of the paper deals only with EMT. It would be of my opinion that the miRNA information should be better integrated into the manuscript or the title should be modified in order to better reflect the data within the paper

R: We appreciate the observation. According to the suggestion of the reviewer we decided to change the title of the paper to better suit the contents of the review. Besides, we consider that is important to clearly assess EMT in our review because is a central mechanism that, according to our hypothesis, could potentially interconnect the expression of miRNAs with expression of genes that are currently established FOLFOX biomarkers. EMT is also a process that is highly correlated with cancer progression and metastasis, which accordingly occurs in the same patients that frequently exhibit higher rates of therapeutic failure and incidence of chemoresistance.

We consider that the novelty of our manuscript resides in exploring the connection between these three phenomena that have not been considered to have a relationship in past research.

  1. Title: Misspelling in title: trough – through

R: The title of the review has been changed. Thus, the misspelling is loss.

  1. Line 57: The use of targeted therapies: The use of targeted therapies is now predominantly first line: cetuximab is used as a first-line agent in patients with Kras wild type and bevacizumab is given in combo with folfox/ folfiri in Kras mutant CRC.

R: We appreciate the observation. The line has been rephrased for better accuracy. Although several countries do provide targeted therapies as first-line CRC treatment, we also wanted to refer to the reality of developing countries as ours, that only consider targeted therapies as second-line treatments because of the economic cost, and often implies a high economic burden for the lower economic class patients.

  1. Line 120: The title of the section is unclear and needs to be reworded

R: The title has been rephrased.

  1. Line 152-53: Reference needed regarding the improvement of therapeutic response with the addition of folinic acid to 5-FU 

R: Thanks, the references have been added at the end of the paragraph, now in line 156 of P.4.

  1. Line 156: The title of the section is unclear and needs to be reworded

R: The title has been rephrased, now in line 157 of P.4.

  1. Line 165: Reference needed

R: Thanks, the references have been added at the end of the paragraph, now in line 166 of P.4.

  1. Pay attention to table numbering. Two table 1 exist.

R: We appreciate this observation. The table number has been corrected in Table 2.

  1. Line 201-206 This statement need to be backed up with references.

R: The reference has been added, now in line 205 P.6, for the first line “Genotyping studies often show inconsistencies regarding the effect of some genetic polymorphisms over relevant outcomes like overall survival, progression-free survival, or risk of adverse events, and sometimes the impact of these associations is marginal”, although the reference is the same as the previous phrase because this statement is extracted from the same publication.

For the next sentence “This suggests that the expression and function of DPYD, TYMS, MTHFR, ERCC1, ERCC2, XRCC1, and GSTP1 may be influenced not only by genetic polymorphisms but also by processes that are specific for the tumor tissue”, this phrase is part of our hypothesis and is currently being assessed as part of our research. However, in order to clarify we have changed a bit the sentence by ““Therefore, the expression and function of DPYD, TYMS, MTHFR, ERCC1, ERCC2, XRCC1, and GSTP1….”

  1. Section 5.1 and 5.2 miRNAs: A discussion here about the clinical relevance of these markers would be appropriate. Most if not all of these studies have been performed in vitro in well established cell lines. It is worth noting that the cell lines are generated to develop resistance rather that multi-clonal populations that exist in a tumour.  Have these miRNAs been shown to contribute to clinical acquired resistance or clinical response? This would add value to the review and interest to the reader. Even an animal study delivering antagomirs would show relevance. 

R: We appreciate this smart observation. A brief paragraph revising some findings regarding the role of miRs and chemoresistance in clinical trials has been added between lines 375-384 of P.13.

  1. A clear link between EMT and the apoptosis regulators is not clear. The section is difficult to follow and it is not certain if the Mir targeting the apoptotic modulators is responsible for the EMT phenotype as well.
  1. We agree with the reviewer and appreciate the observation. The paragraphs have been rephrased for better comprehension, now in lines 311-2, and lines 316-318 of P.11, lines 344, 347, and 349 of P.12, and lines 350-4 of P.13., and the relationship between EMT and the miR has been better explained according to the findings of the cited publication. We hope these changes satisfy the requirements of the reviewer.

Attached you will find the corrected version of the manuscript with changes highlighted in yellow.

Reviewer 2 Report

  1. In line 63 of P.2, “it is important to explore the molecular mechanisms that participate in the development..” could be changed as “it is important to explore the molecular mechanisms involved in the development..”.
  2. In line 134 of P.4, “which is the limiting step in the synthesis of” should be changed as “which is the rate-limiting step in the synthesis of”.
  3. In line 140 of P.4, “into 5-FU metabolites that are integrated into the RNA and DNA” could be changed as “into 5-FU metabolites that are incorporated into the RNA and DNA”.
  4. In lines 218-9 of P.6, “working as a sort of positive feedback between both genes which specific mechanism remains to be explored” could be changed as “working as a sort of positive feedback between both genes where the specific mechanism remains to be explored”.
  5. In line 233 of P.7, “XRCC1 seems to be the only gene that has shown to be downregulated by EMT” should be changed as “XRCC1 seems to be the only gene that has been shown to be downregulated by EMT”.
  6. In lines 375-6 of P.11, “Although the research in the field is scarce, the evidence we exhibit in this review..” could be changed as “Although the research in the field is scarce, the evidence we show (or present) in this review..”.
  7. In line 378 of P.11, “genes that we currently know influence chemotherapy response” should be changed as “genes currently known to influence chemotherapy response”.
  8. In lines 378-9 of P.11, the authors need to be very careful to propose microRNAs as new promising “pharmacogenetic” biomarkers for chemoresistance development” because pharmacogenomics (or pharmacogenetics) usually considers the relationship between an individual’s genetics and their response to medications (i.e., drug-gene interactions) and microRNAs have not yet been considered as an individual’s genetics.     
  9. In line 382 of P.12, “Over the last years, it has been assumed that ..” should be changed as “Over the last few years, it has been assumed that ..”.
  10. In lines 384-5 of P.12, “This phenomenon has been attributed to their reduced proliferative activity and the expression of efflux transporters” should be changed as “This phenomenon has been attributed to their reduced proliferative activity and increased expression of efflux transporters”.
  11. In line 392 of P.12, “Plasticity is one of the characteristics that makes cancer treatment such a difficult task” should be changed as “Cancer cell plasticity is one of the characteristics that makes cancer treatment such a difficult task”.
  12. In line 404 of P.12, “Herein, we have also described evidence that supports ..” should be changed as “Herein, we have also presented evidence that supports ..”.
  13. In line 438 of P.13, the authors need to explain the meaning of “mRNA processivity”?
  14. In line 451 of P.13, “These associations need to be supported by experimental evidence ..” should be changed as “These associations need to be validated by experimental evidence ..”.
  15. In line 459 of P.14, “If the hypothesis proposed in this review is supported by scientific evidence” should be changed as “If the hypothesis proposed in this review is confirmed by scientific evidence”.
  16. In lines 462-3 of P.14, “could proportionate better tools not only to predict FOLFOX response, but also to follow the evolution of FOLFOX sensitivity” should be changed as “could represent better tools not only to predict FOLFOX response, but also to track the evolution of FOLFOX sensitivity”.
  17. In lines 475-6 of P.14, “but by the date” should be changed as “but to date”.

   18. In lines 478-9 of P.14, “by upregulation of DPYD, TYMS, MTHFR, ERCC1,           ERCC2, XRCC1, and 478 GSTP1 genes expression in cancer cells by                 EMT-TFs” should be changed as “by upregulating the expression of                   DPYD, TYMS, MTHFR, ERCC1, ERCC2, XRCC1, and 478 GSTP1 genes in             cancer cells via stimulating EMT-TFs” .

Author Response

We greatly appreciate the pertinent observations of the referee, which have improved considerably our manuscript.   Below you will see the point-by-point response to the comments.  

1. In line 63 of P.2, “it is important to explore the molecular mechanisms that participate in the development..” could be changed as “it is important to explore the molecular mechanisms involved in the development..”.

R: Thanks, the modification has been incorporated in the manuscript, now in line 64 of P.2.

2. In line 134 of P.4, “which is the limiting step in the synthesis of” should be changed as “which is the rate-limiting step in the synthesis of”.

R: Thanks, the modification has been incorporated in the manuscript, now in line 135 of P.4.

3. In line 140 of P.4, “into 5-FU metabolites that are integrated into the RNA and DNA” could be changed as “into 5-FU metabolites that are incorporated into the RNA and DNA”

R: Thanks, the modification has been incorporated in the manuscript, now in lines 141 of P.4.

4. In lines 218-9 of P.6, “working as a sort of positive feedback between both genes which specific mechanism remains to be explored” could be changed as “working as a sort of positive feedback between both genes where the specific mechanism remains to be explored”.

R: Thanks, the modification has been incorporated in the manuscript, now in lines 218-9 of P.7.

5. In line 233 of P.7, “XRCC1 seems to be the only gene that has shown to be downregulated by EMT” should be changed as “XRCC1 seems to be the only gene that has been shown to be downregulated by EMT”.

R: Thanks, the modification has been incorporated in the manuscript, now in line 233 of P.7.

6. In lines 375-6 of P.11, “Although the research in the field is scarce, the evidence we exhibit in this review..” could be changed as “Although the research in the field is scarce, the evidence we show (or present) in this review..”.

R: Thanks, the modification has been incorporated in the manuscript, now in line 392 of P.13.

7. In line 378 of P.11, “genes that we currently know influence chemotherapy response” should be changed as “genes currently known to influence chemotherapy response”.

R: Thanks, the modification has been incorporated in the manuscript, now in line 394 of P.13.

8. In lines 378-9 of P.11, the authors need to be very careful to propose microRNAs as new promising “pharmacogenetic” biomarkers for chemoresistance development” because pharmacogenomics (or pharmacogenetics) usually considers the relationship between an individual’s genetics and their response to medications (i.e., drug-gene interactions) and microRNAs have not yet been considered as an individual’s genetics.     

R: We really appreciate this observation. We removed the word pharmacogenetic in the phrase considering the potential misleading interpretation, but kept pharmacogenomics at the end of the phrase because the proposed mechanism would effectively expand the field and complement the current knowledge of pharmacogenetics and pharmacogenomics.

9. In line 382 of P.12, “Over the last years, it has been assumed that ..” should be changed as “Over the last few years, it has been assumed that ..”.

R: Thanks, the modification has been incorporated in the manuscript, now in line 398 of P.13.

10. In lines 384-5 of P.12, “This phenomenon has been attributed to their reduced proliferative activity and the expression of efflux transporters” should be changed as “This phenomenon has been attributed to their reduced proliferative activity and increased expression of efflux transporters”.

R: Thanks, the modification has been incorporated in the manuscript, now in line 401 of P.13.

11. In line 392 of P.12, “Plasticity is one of the characteristics that makes cancer treatment such a difficult task” should be changed as “Cancer cell plasticity is one of the characteristics that makes cancer treatment such a difficult task”.

R: Thanks, the modification has been incorporated in the manuscript, now in line 408 of P.14.

12. In line 404 of P.12, “Herein, we have also described evidence that supports ..” should be changed as “Herein, we have also presented evidence that supports ..”.

R: Thanks, the modification has been incorporated in the manuscript, now in line 421 of P.14.

13. In line 438 of P.13, the authors need to explain the meaning of “mRNA processivity”?

R: We really appreciate this comment. In order to overcome the observation, we have added a short explanation inside parentheses in lines 455-6 of P.15.

14. In line 451 of P.13, “These associations need to be supported by experimental evidence ..” should be changed as “These associations need to be validated by experimental evidence ..”.

R: Thanks, the modification has been incorporated in the manuscript, now in line 469 of P.15.

15. In line 459 of P.14, “If the hypothesis proposed in this review is supported by scientific evidence” should be changed as “If the hypothesis proposed in this review is confirmed by scientific evidence”.

R: Thanks, the modification has been incorporated in the manuscript, now in line 476 of P.15.

16. In lines 462-3 of P.14, “could proportionate better tools not only to predict FOLFOX response, but also to follow the evolution of FOLFOX sensitivity” should be changed as “could represent better tools not only to predict FOLFOX response, but also to track the evolution of FOLFOX sensitivity”.

R: Thanks, the modification has been incorporated in the manuscript, now in lines 479-80 of P.15.

17. In lines 475-6 of P.14, “but by the date” should be changed as “but to date”.

R: Thanks, the modification has been incorporated in the manuscript, now in line 491 of P.16.

18. In lines 478-9 of P.14, “by upregulation of DPYD, TYMS, MTHFR, ERCC1, ERCC2, XRCC1, and GSTP1 genes expression in cancer cells by EMT-TFs” should be changed as “by upregulating the expression of DPYD, TYMS, MTHFR, ERCC1, ERCC2, XRCC1, and GSTP1 genes in cancer cells via stimulating EMT-TFs”.

R: Thanks, we agree with this change. The modification has been incorporated in the manuscript, now in lines 404-5 of P.16.

Attached you will have the corrected version of the manuscript with changes highlighted in yellow.

This manuscript is a resubmission of an earlier submission. The following is a list of the peer review reports and author responses from that submission.

Round 1

Reviewer 1 Report

This is a review article summarized the potential role of MicorRNAs as epigenetic modulators of FOLFOX chemotherapy response through epithelial-mesenchymal transition in the colorectal cancer. It is quite well organized but there are some comments would like to author to address.

Major concern:

The article included nice background of FOLFOX chemo regimen and the potential biomarker for the chemo compounds. the basics of MicroRNAs. Although the article listed miRNAs that affect 5Fu and oxaliplatin response and the link between EMT and chemo response, I could not see any clear evidence to link the MicroRNAs with chemo response via EMT except one sentence on page 13 line 406 (regarding the role of miR23b). This are missing links between the microRNAs and EMT and chemo response.  Majority of the targeted genets by miRs listed in table 1 and table 2 are not overlapping with the genes listed in table 3. As the authors mentioned in the article (page 13 line 388), the chemoresistance via EMT "may be affected by miRs". It is very hard to let the readers to draw the same conclusion as the authors that miRs could be an epigenetic marker for EMT or chemo response.

Minor concerns:

  1. Title : page 1 line 3 : "trough" should be "through"
  2. Introduction: page 2 line 69-71. The authors may want to reorganize the current treatment overview. For example, some high-risk stage II disease still need adjuvant chemo treatment. Neoadjuvant treatment is not commonly used for stage III colon cancer.
  3. Introduction : page 2 line 79. "other association" may be changed to "other regimens".
  4. Introduction:  Page 2 line 82. "Antiangioenic"  may be changed to "angiogenic".
  5. Introduction page 3 line 84. The conclusion that "targeted therapies are not widely applied" is not true.
  6. Page for line 143. The authors may want to add one more sentence to explain the mechanism of leucovorin.